# Rolling-Translated circRUNX2.2 Promotes Lymphoma Cell Proliferation and Cycle Transition in Marek’s Disease Model

**DOI:** 10.3390/ijms252111486

**Published:** 2024-10-25

**Authors:** Lulu Wang, Gang Zheng, Yuqin Yang, Junfeng Wu, Yushuang Du, Jiahua Chen, Changjun Liu, Yongzhen Liu, Bo Zhang, Hao Zhang, Xuemei Deng, Ling Lian

**Affiliations:** 1Frontiers Science Center for Molecular Design Breeding (MOE), China Agricultural University, Beijing 100193, China; 2MOE Key Laboratory of Bioinformatics, School of Life Sciences, Tsinghua University, Beijing 100084, China; 3Division of Avian Infectious Diseases, Harbin Veterinary Research Institute of Chinese Academy of Agricultural Sciences, Harbin 150001, China

**Keywords:** lymphoma, Marek’s disease, chicken, rolling circle manner, CircRUNX2.2-rt

## Abstract

Marek’s disease (MD), an immunosuppressive disease induced by the Marek’s disease virus (MDV), is regarded as an ideal model for lymphoma research to elucidate oncogenic and anti-oncogene genes. Using this model, we found that circRUNX2.2, derived from exon 6 of RUNX2, was significantly upregulated in MDV-infected tumorous spleens. In this study, we deeply analyzed the potential role of circRUNX2.2 in lymphoma cells. An open reading frame (ORF) in circRUNX2.2 with no stop codon was predicted, and small peptides (named circRUNX2.2-rt) presenting multiple ladder-like bands with different molecular weights encoded by circRUNX2.2 were detected via Western blotting assay. The polysome fraction assay reconfirmed the translation ability of circRUNX2.2, which could be detected in polysome fractions. Subsequent analysis verified that it translated in a rolling circle manner, rather than being assisted by the internal ribosome entry site (IRES) or m6A-mediated mechanism. Furthermore, we found that circRUNX2.2-rt was potently induced in MSB1 cells treated with sodium butyrate (NaB), which reactivated MDV and forced the MDV transition from the latent to reactivation phase. During this phase, MDV particles were clearly observed by electron microscopy, and the viral gene pp38 was also significantly upregulated. A biological function study showed that circRUNX2.2-rt promoted cell proliferation and cell cycle transition from the S to G2 phase and inhibited the apoptosis of MSB1. Further immunoprecipitation and mass spectrometry assays showed that 168 proteins potentially interacting with circRUNX2.2-rt were involved in multiple pathways related to cell cycle regulation, which proved that circRUNX2.2-rt could bind or recruit proteins to mediate the cell cycle.

## 1. Introduction

Lymphoma, a frequently diagnosed form of lymphoid malignancy, is in the top ten of most common cancers worldwide. Despite advancements in targeted therapeutic approaches that have led to successful outcomes in some individuals with lymphoma, the management of refractory and relapsed cases remains a big challenge, highlighting the need for further investigation of lymphoma [1]. Marek’s disease (MD) is an infectious lymphoproliferative disease caused by Marek’s disease virus (MDV) in chickens, characterized by rapid-onset lymphomas in multiple organs [2]. MD provides an ideal model for studying the molecular mechanisms and therapeutic targets of lymphoma, and greatly facilitates the in-depth study of lymphoma and the development of clinical applications. In the poultry industry, the outbreaks of MD caused huge losses, and the main approach to control MD was the subcutaneous injection of the MD vaccine into one-day-old chicks. The costs of health interventions were more than USD 1 billion per year. However, the use of vaccines has led to the continuous evolution of viruses towards higher virulence [3], and analysis of the ancient and modern MDV genomes showed that the virulence of ancient MDV strains was much lower than that of modern strains [4]. Therefore, researchers have attempted to analyze host genetic resistance, susceptibility factors and viral infection mechanisms, aiming to better prevent and control the disease as well as to contribute to lymphoma research.

Circular RNA (circRNA) is a type of noncoding RNA with a closed loop structure formed by ligating a downstream splice donor site to an upstream splice acceptor site during pre-mRNA splicing. CircRNA biogenesis studies have revealed that back-splicing circularization was far less favorable than canonical splicing. Unlike linear mRNA, circRNA was highly stable and resistant to exonuclease-mediated degradation due to its covalently closed circular structure and could accumulate to realize high abundance [5]. Based on its characteristics, circRNA has been reported to be involved in biological processes through multiple mechanisms of action, including miRNA sponges [6], alternative splicing and transcriptional regulation [7,8], protein interaction [9] and translation [10]. To date, several studies have reported that circRNAs exerted important regulatory roles in diseases and individual development in humans and animals [11], and only a few studies have uncovered circRNAs involved in the development of MD [12,13,14].

In this study, a significantly upregulated circRNA in MDV-infected tumorous spleen tissues, circRUNX2.2, was selected from previous data [14,15], and its role in MD development was further investigated. circRUNX2.2 was generated from exon 6 of the RUNX2 gene, a member of the Runt domain family of transcription factors that was involved in the development of skeleton [16] and malignant tumors [17]. We analyzed its potential roles during MDV infection, which will contribute to a deeper understanding of the mechanisms of host and MDV interactions.

## 2. Results

### 2.1. CircRUNX2.2 Possessed Translation Ability

The open reading frame (ORF) of circRUNX2.2 was predicted using Open Reading Frame Finder (https://www.ncbi.nlm.nih.gov/orffinder/, accessed on 5 November 2021), and only one ORF without a stop codon, which started at nucleotide 69th, was predicted. To test the translation potential of the predicted ORF, fusion expression vectors of the red fluorescent protein (RFP) and ORF were constructed. The entire coding sequence (CDS) of RFP was amplified from the pCDH-CMV-MCS-EF1-RFP-T2A-Puro vector and cloned into the pcl-ciR5 vector to form an RFP-positive expression vector named pcl-ciR5-RFP (Figure 1A). Meanwhile, RFP CDS with start codon ATG deletion was cloned into the pcl-ciR5 vector to construct an RFP negative expression vector named pcl-ciR5-RFP-ΔATG (Figure 1B). The above-two RFP expression vectors were transfected into DF-1 cells to observe fluorescent protein signals. The results showed that fluorescent signals of vector-containing GFP were observed in both groups (Figure 1A,B), indicating that both vectors were successfully transfected into DF-1 cells. The RFP fluorescent signals were observed only in the pcl-ciR5-RFP group (Figure 1A), indicating that the positive and negative vectors worked well. The negative control vector could be used to investigate the translation capacity by recombining with target sequences. The complete ORF sequence of circRUNX2.2 were cloned into pcl-ciR5-RFP-ΔATG to construct the pcl-ciR5-circRUNX2.2-RFP-ΔATG vector and transfected into DF-1 cells. The results showed that both green and red fluorescence were observed (Figure 1C), which suggested that RFP could be expressed with the help of ORF sequences of circRUNX2.2 when the start codon ATG of RFP was deleted. Furthermore, circRUNX2.2 ORF sequences with start codon ATG deletion were cloned into pcl-ciR5-RFP-ΔATG to construct the pcl-ciR5-circRUNX2.2-ΔATG-RFP-ΔATG vector (Figure 1D), which was transfected into DF-1 cells, and there was no red fluorescence observed. These results suggested circRUNX2.2 had translation ability.

To further confirm the translatability of circRUNX2.2, we examined its distribution in different ribosomal fractions in MSB1 cells. The sucrose buffer (10% and 50%) was prepared for gradient centrifugation, and 40S subunit, 60S subunit, 80S monosome and polysome-bound RNA were isolated at different densities via ultracentrifugation. RNA in each fraction was extracted, and its expression was measured via q-PCR. The results showed that the positive control genes GAPDH and β-actin were mainly distributed in polysome fractions, and circRUNX2.2 was also distributed in polysome fractions, which confirmed its ability to bind ribosomes and its translation ability (Figure 1E).

### 2.2. The Translation of circRUNX2.2 Was Independent of Internal Ribosome Entry Site (IRES) and m^6^A Modification

Internal ribosome entry site (IRES) and m^6^A RNA modification were known to contribute to circRNAs’ translation. To evaluate the mechanism of cap-independent translation of circRUNX2.2, IRES and m^6^A sites in circRUNX2.2 were predicted. The predicted results indicated that circRUNX2.2 contained two IRES segments which were located at 1–18 nt and 120–129 nt, respectively. Using the pcl-ciR5-circRUNX2.2-RFP-ΔATG vector as a template, the predicted fragments of IRES1 and IRES2 were respectively deleted to construct the pcl-ciR5-circRUNX2.2-RFP-ΔATG-ΔIRES1 and pcl-ciR5-circRUNX2.2-RFP-ΔATG-ΔIRES2 vectors to detect the activity of IRES (Figure 2A,B). The results showed that DF-1 cells transfected with the pcl-ciR5-circRUNX2.2-RFP-ΔATG-ΔIRES1 vector and pcl-ciR5-circRUNX2.2-RFP-ΔATG-ΔIRES2 vector could still detect red fluorescence (Figure 2A,B), suggesting that deletion of either IRES1 or IRES2 did not affect the translation of circRUNX2.2. Interestingly, a predicted m^6^A modification site was at nucleotide 125th which happened to locate in the IRES2 region (Figure 2B). As mentioned above, red fluorescence was detected when the IRES2 region was deleted; hence, it suggests that the translation of circRUNX2 was not mediated by m^6^A modification.

To avoid any missing IRESs due to prediction limitations, circRUNX2.2 whole sequences were repeated twice and then cloned into the Luc2-IRES-Report vector to construct the Luc2-IRES-circRUNX2.2 vector (Figure 2C), and IRES activity was detected by measuring the luciferase activity of Luc relative to that of RLuc. The results showed that 293T cells transfected with the positive control plasmid Luc2-EMCV-IRES-Report showed the highest luciferase activity (Luc/Rluc), while the cells transfected with Luc2-IRES-circRUNX2.2 vector possessed low luciferase activity, which showed no difference with that transfected with empty vector Luc2-IRES-Report, confirming that circRUNX2.2 did not induce ribosome entry (Figure 2D). These results showed that the translation of circRUNX2 could not be initiated by IRESs or m^6^A modification.

### 2.3. circRUNX2.2 Encoded Small Peptide in a Rolling Circle Translation Manner

Some previous studies have reported that circRNAs could be translated in a manner similar to rolling circle amplification of polymerase reaction when their length was a multiple of three and they had no stop codon [18,19,20,21,22]. Our bioinformatic analysis by Open Reading Frame Finder (ORF) showed that circRUNX2.2 had one ORF without a stop codon, and its total number of nucleotides was exactly 162 nt. Thus, we suspected that circRUNX2.2 could be translated into a protein in a rolling circle translation manner. To test this hypothesis, the RFP CDS sequence without start codon (ATG) and stop codon (TAA) were cloned into the pcl-ciR5-circRUNX2.2 vector to construct the pcl-ciR5-circRUNX2.2-RFP-ΔATG-ΔTAA-17aa vector, in which the RFP was located at 17 amino acids (aa) downstream of the ATG of circRUNX2.2 (Figure 3A). Consistent with the results of the pcl-ciR5-circRUNX2.2-RFP-ΔATG vector treatment, red fluorescence was highly expressed in DF-1 cells transfected with the pcl-ciR5-circRUNX2.2-RFP-ΔATG-ΔTAA-17aa vector. Then, 3×FLAG sequences were cloned into the pcl-ciR5-circRUNX2.2 vector to construct the pcl-ciR5-circRUNX2.2-FLAG-17aa vector (Figure 3B). Western blotting with anti-FLAG showed that ladder bands with different molecular weights were detected (Figure 3C), which was consistent with the characteristics of protein rolling circle translation [19,20]. The results showed that the ~15 kDa band was the main product, indicating that circRUNX2.2 was translated almost twice, based on a product of 7.7 kDa translated from the linear counterpart of the circular RNA (Figure 3B). These results revealed that circRUNX2.2 could be translated in a rolling circle manner.

To access the conservation of circRUNX2.2, its sequences derived by exon 6 of the RUNX2 gene from different species sources, included chicken, human, rat, cat and pig, were downloaded from the circAtlas 3.0 (https://ngdc.cncb.ac.cn/circatlas/, accessed on 6 July 2023). Their lengths were all 162nt and had high sequence similarity (>80%) (Appendix A). The ORF of these different species were predicted to investigate their encoding capacity, and the results showed that their ORF were all at the same position. The predicted cyclic polypeptide sequences among species showed only one amino acid difference between rat and the other four species, and one amino acid difference between chicken and the other four species (Appendix A), which indicated high cross-species conservation of amino acid sequences from different species. Furthermore, we overexpressed human circRUNX2.2-FLAG in 293T cells, and the FLAG proteins with different molecular weights showing ladder-like bands were also detected (Appendix A). Additionally, the FLAG signaling was detected regardless of whether FLAG was located at the N-terminus or C-terminus of circRNAs by IF assay, which was consistent with that we found in chickens (Appendix A). It suggested that circRUNX2.2 small peptide could be efficiently translated from different species sources, which indicated similar biological function of circRUNX2.2 across species.

To analyze the translation product of circRUNX2.2, we firstly predicted the sequences of circRUNX2.2- and FLAG-fused proteins (Figure 3D). Subsequently, the pcl-ciR5-circRUNX2.2-FLAG-17aa vector was transfected into DF-1 cells, and FLAG magnetic beads were used to enrich the circRUNX2.2_FLAG fused translation product for mass spectrometry assay. The sequences of the translation products (Figure 3E–G) were consistent with the predicted sequences. These data further confirmed that circRUNX2.2 could be translated in a rolling circle manner.

### 2.4. The Protein Product of pcl-ciR5-circRUNX2.2-RFP-ΔATG-ΔTAA Was Originated from Circular Not Linear Transcripts

Since RFP or FLAG sequences were both constructed at 17aa downstream of ATG in circRUNX2.2 ORF, which was upstream of reverse-cyclization-mediating sequences (Figure 3A,B), the linear expression products of RFP or FLAG might also be produced at the same time. To confirm that the fluorescence detected in our experiments was from the cyclic transcription product rather than the linear product, three additional vectors, pcl-ciR5-circRUNX2.2-RFP-ΔATG-ΔTAA-29aa, -41aa and 54aa, were constructed. For the first vector, RFP was located at 29aa downstream of ATG in circRUNX2.2 ORF and upstream of reverse-cyclization-mediating sequences (Figure 4A); for the second and third vector, RFP were respectively located at 41aa (Figure 4B) and 54aa (Figure 4C) downstream of ATG in circRUNX2.2 ORF and the downstream of reverse-cyclization-mediating sequences. The results showed that cells treated with any of the three vectors exhibited red fluorescence. The pcl-ciR5-circRUNX2.2-RFP-ΔATG-ΔTAA-29aa vector showed a red fluorescent signal as high as those transfected with the pcl-ciR5-circRUNX2.2-RFP-ΔATG-ΔTAA-17aa vector (Figure 3A and Figure 4A), whereas a weak red fluorescent signal was observed in cells transfected with the pcl-ciR5-circRUNX2.2-RFP-ΔATG-ΔTAA-41aa and pcl-ciR5-circRUNX2.2-RFP-ΔATG-ΔTAA-54aa vectors (Figure 4B,C). There were two possibilities for the results: (1) The signals from pcl-ciR5-circRUNX2.2-RFP-ΔATG-ΔTAA-17aa and -29aa were mainly from linear products, which showed a significantly bright fluorescence compared to -41aa and -54aa, which could not produce linear products due to the insertion position of RFP downstream of the reverse-cyclization-mediating sequence. (2) The different signal intensity between -17aa, -29aa vector and the other two vectors (-41aa or 54aa) resulted from the effect of different distances between the inserted fragments and the ATG of circRUNX2.2, rather than the linear product. To determine this, the reverse-cyclization-mediating sequences in pcl-ciR5-circRUNX2.2-RFP-ΔATG-ΔTAA-29aa and pcl-ciR5-circRUNX2.2-RFP-ΔATG-ΔTAA-54aa were deleted to block the generation of circular RNA to construct pcl-ciR5-circRUNX2.2-RFP-ΔATG-ΔTAA-29aa-ΔR/G and pcl-ciR5-circRUNX2.2-RFP-ΔATG-ΔTAA-54aa-ΔR/G vectors (Figure 4D,E). A slight RFP signal was detected in pcl-ciR5-circRUNX2.2-RFP-ΔATG-ΔTAA-29aa-ΔR/G, and no RFP signal was detected in pcl-ciR5-circRUNX2.2-RFP-ΔATG-ΔTAA-54aa-ΔR/G. These results illustrated that high red fluorescent protein signals observed in pcl-ciR5-circRUNX2.2-RFP-ΔATG-ΔTAA-29aa were mainly expressed from circular RFP; although the signal of linear RFP expression was present, it was very weak. Further results showed that all expressed RFP proteins were located in the cytoplasm (Appendix A).

### 2.5. Rolling Circle Translation Manner and Ladder-like Small Peptide Products Derived from circRUNX2.2 Were Reconfirmed

The RFP CDS sequences were replaced with 3×FLAG sequences to construct pcl-ciR5-circRUNX2.2-3×FLAG-29aa, pcl-ciR5-circRUNX2.2-3×FLAG-41aa and pcl-ciR5-circRUNX2.2-3×FLAG-54aa vectors (Figure 5A–D). Western blot assay with anti-FLAG also detected different molecular bands in these four groups; the ~15 kDa band was the main product (Figure 5E), and as the distance between FLAG and the start codon ATG increased, the detected ladder bands became increasingly weaker. Immunofluorescence with an anti-FLAG assay was performed to detect the fused FLAG protein expression (Figure 5F). The results showed that the fused protein was located in the cytoplasm (Appendix A), and the effect of different locations of 3×FLAG on the translation efficiency was also observed. Hence, we reconfirmed that ladder-like small peptide products were produced from circRUNX2.2 in a rolling circle translation manner and that the change in RFP and FLAG translation efficiency was related to the distance of their insertion site from the start codon ATG.

### 2.6. circRUNX2.2 Expression Was Induced During MDV Reactivation

MSB1, a well-known lymphoblastoid cell line (LCL) derived from MD lymphomas, has served as a valuable model for investigating the features of transformed cells. In this study, we chose two MSB1 cell states to study the function of circRUNX2.2. One was wild-type MSB1 cells, wherein MDV was latent, and the other was MSB1 treated with sodium butyrate (NaB), which could switch MDV from latency to lytics [23,24,25]. Here, we treated MSB1 cells with NaB (agonist) for 48h, and the MDV-encoded phosphoprotein pp38, which is strongly associated with lytic replication, was significantly upregulated (Figure 6A), and viral particles were detected, which were mainly distributed in the nucleus (Figure 6B). Interestingly, the endogenous transcriptional expression of circRUNX2.2 was also upregulated during this phase (Figure 6A). The significant increase in the circRUNX2.2 main product (the encoded product of circRUNX2.2) underwent two rounds and its molecular weight was approximately 10 kDa, detected by Western blot assay, using an antibody in which the antigenic sequence was precisely located within the peptide segment encoded by exon 6 (the source of circRUNX2.2) of the RUNX2 gene (Figure 6C). These results indicated that circRUNX2.2 was involved in the transition of MSB1 from latency to reactivation.

### 2.7. circRUNX2.2 Promoted MSB1 Cell Proliferation and Cell Cycle Progression from S to G2 Phase and Reduced the Apoptosis via Encoded Protein

To analyze the effects of circRUNX2.2 on wild-type MSB1, overexpression vectors with or without ATG (ov_circRUNX2.2, ov_circRUNX2.2-ΔATG) were constructed and transfected into MSB1 cells. The results of CCK-8 and EdU assays showed that MSB1 cell proliferation was significantly promoted in the ov_circRUNX2.2-treated group compared to the control group (pcl-ciR5) (*p* < 0.05), whereas ov_circRUNX2.2-ΔATG had no significant effect on MSB1 cell proliferation (*p* > 0.05) (Figure 7A–C). The cell cycle assay revealed that ov_circRUNX2.2 significantly promoted the transition of MSB1 cells from the S to G2 phase (*p* < 0.05) (Figure 7D). The apoptosis assay results revealed that ov_circRUNX2.2 inhibited apoptosis (*p* < 0.05) (Figure 7E). Taken together, these data demonstrated that circRUNX2.2-rt promotes MSB1 cell development.

To further elucidate the regulatory network of circRUNX2.2 in the promoting of lymphoma cell development, proteins potentially interacting with circRUNX2.2-rt were explored. The pcl-ciR5-circRUNX2.2-3×FLAG-17aa vector was transfected into MSB1 cells, and FLAG magnetic beads were used to enrich circRUNX2.2-FLAG and its interacting proteins. In total, 168 proteins were detected via mass spectrometry (Appendix A) and submitted to the STRING (version 12.0) database (https://cn.string-db.org/, accessed on 26 July 2023) for functional analysis with default parameters [26]. The reactome channel in the STRING database was used to conduct the pathway analysis. The results showed that the enriched proteins were involved in the regulation of various pathways, especially the cell cycle (Figure 7F). Therefore, we considered that circRUNX2.2-rt promotes MD by interacting with proteins involved in cell cycle regulation to promote cell proliferation, accelerate the transition from the S to G2 phase, and inhibit cell apoptosis, which indicated the oncogenic function of circRUNX2.2-rt.

To further elucidate the potential regulatory network of circRUNX2.2 in promoting lymphoma cell proliferation and cell cycle progression from S to G2 phase and reducing the apoptosis, we conducted RNA sequencing (RNA-seq) in MSB1 cells with circRUNX2.2, circRUNX2.2-ΔATG and pcl-ciR5 overexpression, respectively. In total, 1533, 1364 and 1336 differentially expressed genes were identified in three comparison groups (circRUNX2.2_vs._pcl-ciR5, circRUNX2.2-ΔATG_vs._pcl-ciR5 and circRUNX2.2_vs._circRUNX2.2-ΔATG), respectively (Appendix A). The gene expression trends among three groups were mainly clustered into eight clusters, among which, cluster1, cluster3 and cluster5 were significantly enriched (*p* < 0.05) (Appendix A). Gene expression trends analysis showed that genes enriched in cluster 5 and cluster 2 were upregulated and downregulated, respectively, in circRUNX2.2 treatment group compared with circRUNX2.2-ΔATG and pcl-ciR5 treatment groups. From these two clusters, 40 significantly upregulated genes and 77 significantly downregulated genes were identified for further enrichment analysis (Appendix A). GO enrichment analysis showed that 40 upregulated DEGs were significantly enriched in GO terms related to protease activity (Appendix A), and KEGG pathways related to cellular processes such as the cell cycle, autophagy, apoptosis and necrosis (Appendix A). Notably, ubiquitin-specific peptidase 7 (USP7), a hydrolase that deubiquitinates target proteins, was significantly upregulated in the circRUNX2.2 treatment group. The 77 downregulated DEGs were significantly enriched in GO terms related to ubiquitin-associated protease activity (Appendix A), and KEGG pathways related to cellular processes such as the autophagy and necrosis, and viral diseases-related processes such herpes simplex virus 1 infection (Appendix A). Collectively, the results suggested that regulation of protein ubiquitination/deubiquitination was involved in circRUNX2.2′s promotion on lymphoma cell development.

## 3. Discussion

CircRNAs, products of pre-mRNA backing-splicing, are involved in various diseases [27]. As noncoding RNAs, an increasing number of studies have reported that circRNAs can be translated into peptides to participate in the occurrence and development of diseases. For instance, 87 amino-acid peptides encoded by the circular form of the long intergenic non-protein-coding RNA p53-induced transcript (LINC-PINT) could inhibit glioblastoma cell proliferation [28]. Exons 3–4 of FBXW7 encoded 185aa, which inhibited glioma cell proliferation and cell cycle acceleration [29]. A functional protein encoded by circPPP1R12A promotes colon cancer growth and metastasis by activating the Hippo-YAP signaling pathway [30]. A novel protein, SMO-193a. a., encoded by circ-SMO was critical for Hedgehog signaling, driving glioblastoma tumorigenesis, and acted as a novel target for glioblastoma treatment [31]. In the present study, we investigated the function of circRUNX2.2, which was significantly upregulated in MDV-infected tumorous spleen tissues (LFC = 5.58, Padj = 0.74 × 10^−2^) based on previous circRNA expression profiling data. circRUNX2.2 was generated from exon 6 of the RUNX2 gene, a member of the Runt domain family of transcription factors, and played crucial roles in osteoblast development [32] and cancers [33,34]. Our study revealed that circRUNX2.2 could encode small peptides with different molecular weights, and WB assay showed that the molecular weight distribution of these peptides exhibited ladder-like bands.

Due to the lack of a 5′ cap and poly-A tail, the mechanism of circRNA translation was worth exploring. Some studies had reported that circRNAs containing ORFs [29], IRES elements [19], or m6A modified nucleotide sequences [35] could be translated. Besides that, a new translation model “rolling circle manner” was reported, and these circRNAs usually lack stop codon and the number of nucleotides were a multiple of three [19,20]. In vitro synthesis of circRNA of 258nt contained 8×FLAG sequences and high-molecular weight peptides (more than 300 kDa) were detected using a cell-free rabbit reticulocyte lysate system, while the molecular weight of the product encoded by linear 258nt containing 8×FLAG sequences was smaller than 25 kDa [20]. circ-EGFR was formed from exons 14 and 15 of EGFR and obtained an infinite open reading frame that started with ATG but had no in-frame stop codon. The circ-EGFR-Flag vector was transfected into 293T cells and several “ladder-shaped” bands were observed via Western blotting. Rolling-translated circEGFR binds to and stabilizes EGFR and correlates with glioblastoma tumorigenesis and nimotuzumab drug sensitivity [27]. In this study, we identified the first chicken circRNA that encoded small peptides in a rolling-circle manner.

Recently, an increasing number of circRNAs were found to be involved in the development of multiple diseases [36]. circRNAs participated in disease regulation via different functions, such as acting as miRNA sponges [17] and regulating parental gene expression [37]. CircHIPK3 was highly expressed in nasopharyngeal carcinoma (NPC), and represses NPC development via sponging miR-4288 to promote ELF3 expression [38]. CircANKS1B was upregulated in triple-negative breast cancer and acted as a miR-148a-3p and miR-152-3p sponge to increase the expression of transcription factors USF1 and TGF-β1 to activate the TGF-β1/Smad signaling pathway and epithelial-to-mesenchymal transition (EMT) [39]. However, there are few studies on circRNAs in lymphoma, and we investigated the role of circRUNX2.2 in lymphoma using MD as a model. MD is an immunosuppressive disease caused by MDV, a double-stranded DNA virus with a repetitive structure characteristic of the Alphaherpesviridae family [40]. MDV infection could be divided into four main stages in susceptible individuals: (1) early cytolytic phase, (2) latent phase, (3) late cytolytic and immunosuppressive phase and (4) proliferative phase [41]. MSB1 cells provided an ideal model to study the latent infection phase because they contain MDV, but few viral genes were expressed [24,42]. In this study, we found that circRUNX2.2-rt promoted MSB1 cell proliferation and cell cycle transition from the S to G2 phase and inhibited apoptosis. MSB1 cells were treated with sodium butyrate (NaB) to reactivate MDV, and the expression of viral genes [25,43,44] and pp38, an indicator of MDV reactivation, could be induced [45]. Consistent with previous findings [46], we observed MDV particles in MSB1 cells treated with NaB using electron microscopy. The circRUNX2.2 RNA expression and circRUNX2.2-rt expression were also significantly increased. Combining the high expression of circRUNX2.2 in tumor tissues and MSB1 cells with or without MDV reactivation, as well as its promoting effects on the growth of MSB1 cells, we speculated circRUNX2.2 acted as a promoter in MD progression.

Studies on the mechanism of circRUNX2.2 involvement in MD development revealed that circRUNX2.2-rt might interact with multiple proteins, which were involved in the regulation of signaling pathways. Notably, multiple pathways related to cell cycle regulation were enriched, such as the regulation of PLK1 activity at G2/M transition, and G1/S transition. The cell cycle could be divided into four major phases: gap 1 (G1), DNA synthesis (S), gap 2 (G2) and mitosis (M). The aberration of cell cycle progression was one of the basic mechanisms of tumorigenesis [47]. circRNAs have been reported to be involved in the pathogenesis of multiple diseases by regulating the cell cycle [48]. CircTP63, highly expressed in lung squamous cell carcinoma (LUSC) tissues and facilitated cell cycle progression by acting as an miR-873-3p sponge to upregulate the expression of FOXM1, CENPA and CENPB [49]. circNOL10, which is expressed at low levels in lung cancer, inhibited lung cancer cell viability, promoted apoptosis, shorten the S and G2/M phases and inhibited cell proliferation by regulating the HN polypeptide family [50]. The functional protein circPPP1R12A-73aa encoded by circPPP1R12A promoted DLD-1 and Caco-2 cell proliferation, including colony formation and cell cycle, via the Hippo-YAP signaling pathway [30]. In the present study, we found circRUNX2.2-rt interacted proteins were involved in cell cycle regulation, which was consistent with the fact that circRUNX2.2-rt accelerated the transition from the S to G2 phase. Hence, we considered that circRUNX2.2 promoted MSB1 cell proliferation and inhibited MSB1 cell apoptosis by interacting with proteins that participate in cell cycle regulation.

This study identified a novel protein, circRUNX2.2-rt, encoded by circRUNX2.2, a highly expressed circRNA in MDV-infected tumor spleen tissue. The specific translation mechanism analysis revealed that circRUNX2.2 encoded small peptides in a rolling circle manner independent of the IRES and m6A modification mechanism. MSB1 cells were treated with NaB to characterize the progression of MDV from latency to reactivation; MDV virus gene pp38 and circRUNX2.2 expression were both activated, which may contribute to the replication, propagation and tumor transformation of activated viruses in vivo. Functional studies showed that circRUNX2.2-rt might interact with cycle regulation-related proteins to promote MSB1 cell proliferation and cell cycle transition from the S to G2 phase and inhibit cell apoptosis. Currently, several therapeutic strategies were explored with the aim to overexpress or knockdown circRNAs [51]. Considering that circRUNX2.2 functioned as an oncogene in MD, knocking down its expression was the ideal therapeutic strategy. Due to the high GC content at the junction region of circRUNX2.2, it was difficult to design an effective siRNA to knock it down; the CRISPR/Cas9-mediated circRNA knockout or knockdown [52] and the CRISPR/Cas13-mediated circRNA knockdown technologies [53] may contribute to exploring the therapeutics targeting circRUNX2.2.

Ubiquitylation, a post-translational modification, enables mechanistically diverse, quantitative and reversible regulation. Deregulation of ubiquitin signaling was closely associated with various human pathologies [54]. A study reported that lncRNA INKILN was induced in inflammation, which inhibited the MKL1 protein ubiquitin proteasome degradation via interacting with USP10 and enhanced the interaction of MKL1 and p65 in nuclear and subsequent transactivation program of the proinflammatory genes [55]. circMTCL1, upregulated in the advanced laryngeal squamous cell carcinoma (LSCC), could recruit C1QBP protein and inhibit its ubiquitin-proteasome-mediated degradation to exert oncogenic biological characteristics by promoting cell proliferative capability and invasive and migrative abilities [56]. FBW7, a component of ubiquitin ligase SCF complex, targeted several oncoproteins, like cyclin E, MYC, JUN, Notch 1 and Notch 4 [57]. Our study revealed that downregulated genes in the MSB1 cell with circRUNX2.2-rt treatment were involved in protein deubiquitination pathways, and USP7, a gene that related to ubiquitination pathway, was significantly upregulated in genes after circRUNX2.2-rt treatment. Therefore, we speculated that circRUNX2.2-rt promoted lymphoma cell proliferation and cell cycle progression from S to G2 phase and reduced the apoptosis by altering intracellular protein ubiquitination homeostasis.

This study revealed the critical role of circRUNX2.2 and its encoded circRUNX2.2-rt during the MDV latent infection and reactivation stages, which provided insights for understanding the mechanism of tumorigenesis and exploring novel therapeutic strategies. In future study, we will aim to validate the proteins interacting with circRUNX2.2, investigate the effect of them on the development of MD, and resolve the specific molecular mechanisms, to explore therapeutic implications of targeting circRUNX2.2 or pathways for cancer treatment.

## 4. Materials and Methods

### 4.1. Cell Culture

The chicken lymphoblastoid MDCC-MSB1 cells derived from MD lymphomas were maintained in RPMI 1640 medium (Gibco, New York, NY, USA) supplemented with 10% FBS (Gibco, New York, NY, USA) and 1% Penicillin and Streptomycin (Gibco, New York, NY, USA). DF-1 and 293T cells were cultured in Dulbecco’s modified DMEM medium (DMEM; Gibco, New York, NY, USA) containing 10% FBS. All cells were incubated at 37 °C in a humidified chamber with 5% CO_2_. The cell lines used in this study were obtained from our laboratory’s cell bank.

### 4.2. RNA Isolation

Total RNA was extracted using TRIZOL reagent (Invitrogen, Carlsbad, CA, USA) according to the manufacturer’s instructions. RNA degradation and contamination were detected using 1% agarose gel electrophoresis. NanoDrop 2000 (Life Technologies, Carlsbad, CA, USA) was used to measure RNA concentration and purity.

### 4.3. Real-Time qPCR Analysis

Real-time qPCR was performed as previously described [14]. Briefly, a Circular RNA Fluorescence Quantitative PCR kit (Geneseed, Guangzhou, China) was used to detect the gene expression. The β-actin gene was used as a reference gene to determine the relative expression of the genes. The relative expression of the circRNAs was calculated using 2^−ΔΔCt^ method. The primer sequences are listed in Appendix A.

### 4.4. NaB Treatment

MSB1 cells were treated with sodium butyrate (NaB, Sigma-Aldrich, St. Louis, MO, USA), a histone deacetylase inhibitor, to reactivate MDV at a final concentration of 2.5 mM. After 48 h of treatment, RNA and proteins were collected to detect gene expression, and viral particles were visualized by electron microscopy.

### 4.5. Translation Ability Analysis of circRUNX2.2

To investigate the translation ability of circRUNX2.2, circRNA sequences were repeated three times and then submitted to the online site Open Reading Frame Finder (https://www.ncbi.nlm.nih.gov/orffinder/, accessed on 5 November 2021) to predict ORF. Multiple ORF and red fluorescent protein (RFP) fusion expression vectors were constructed to validate the translation capacity of the predicted ORFs. DF-1 cells (2.5 × 10^5^ cells/well) were seeded in a 24-well plate. Constructed vectors were transfected into DF-1 cells, and after 48 h, the translation ability of the ORF was evaluated according to the expression of RFP detected by fluorescence microscopy.

### 4.6. Polysome Fraction Assay

The MSB1 cell polysome fraction assay was conducted as described previously [56]. Briefly, MSB1 cells were treated with 100 µg/mL CHX for 5 min and washed thrice with cold PBS supplemented with 100 µg/mL CHX. Cells were harvested and lysed with 400 µL lysis buffer containing 20 mM Tris (pH 7.5), 150 mM NaCl, 5 mM MgCl2, 1 mM DTT, 1% Triton X-100, 0.1 mg/mL cycloheximide and 100 U/mL RNase inhibitor for 10 min on ice. The polysome lysate was centrifuged at 12,000× *g* for 5 min at 4 °C to pellet the nuclei, and the supernatant was collected and overlaid onto 10–50% (*w*/*v*) sucrose density gradient, followed by ultracentrifugation at 36,000 rpm for 2 h in a Beckman SW41Ti rotor. After centrifugation, 1 mL/min fractions were collected in 15 tubes per gradient and immediately transferred to an ice bucket. The RNA amount in each fraction was analyzed using a 260 nm spectrophotometer within the optical unit of the fractionation system, and a chart displaying the polysome profile of the gradient was drawn. RNA was extracted from fractions using TRIzol LS Reagent (Invitrogen, Carlsbad, CA, USA), and RT-qPCR was conducted to detect circRUNX2.2 expression; GAPDH and β-actin mRNA levels were selected as positive controls.

### 4.7. Translation Mechanism Analysis

CircRNA sequences were repeated twice and submitted to the IRESite database (http://www.iresite.org/, accessed on 18 March 2019) to analyze the Internal Ribosome Entry Site (IRES) segments. CircRNA sequences with the IRES segment were deleted and RFP fusion expression vectors were constructed to validate IRES activity of the predicted IRES segment. Constructed vectors were transfected into DF-1 cells and fluorescence microscopy was used to observe RFP expression to investigate IRES activity. The dual luciferase reporter assay was used to detect the IRES activity of the circRNA sequences. Briefly, circRNA sequences were repeated twice and cloned into the Luc2-IRES-Report vector, which was cloned into the Luc2-IRES-Report vector as a positive control (Luc2-EMCV-IRES-Report). Constructed vectors were transfected into 293T cells for 48 h, and Renilla Luciferase and Firefly Luciferase activity were measured using the Dual-Glo^®^ Luciferase Assay System (Promega, Madison, WI, USA). CircRNA sequences were repeated twice and submitted to SRAMP (http://www.cuilab.cn/sramp, accessed on 20 February 2016) to predict N6-methyladenosine (m6A) modification site. CircRNA sequences with m6A deletion and RFP fusion expression vectors were constructed to validate the ability that m6A mediated translation. Constructed vectors were transfected into DF-1 cells and fluorescence microscopy was used to observe RFP expression to investigate m6A activity. Constructed vectors were also transfected into cells using Lipofectamine 3000 (Invitrogen, Carlsbad, CA, USA), and RFP expression was observed by fluorescence microscopy.

### 4.8. LC-MS/MS

Proteins were digested with trypsin at 37 °C overnight with shaking. The digested peptides were analyzed using a QExactive mass spectrometer (Thermo Fisher, Carlsbad, CA, USA). The mass spectrum data were retrieved using ProteinPilot (V4.5) and Paragon was used as the Uniprot database (https://www.uniprot.org/, accessed on 2 October 2024) retrieval algorithm. The Gallus proteome reference database was used.

### 4.9. Western Blotting Assay

Proteins were extracted using cell lysis buffer for Western blotting and IP (Beyotime, Haimen, China) with phenylmethanesulfonyl fluoride (Beyotime, Shanghai, China) and quantified using the BCA Protein Assay Kit (Beyotime, Shanghai, China). An amount of 20–30 µg of cellular protein was loaded for detection. PVDF membranes (Bio-Rad, Hercules, CA, USA) were used to transfer proteins and blocked with 5% fat-free milk. The primary antibodies against FLAG (ab205606, Abcam, Cambridge, UK), β-actin (ab8227, Abcam, Cambridge, UK) were incubated with membrane overnight at 4 °C. The secondary antibody (Goat Anti-Rabbit antibodies (MF093, Mei5bio, Beijing, China) or Goat Anti-Mouse antibodies (MF094, Mei5bio, Beijing, China) was incubated at room temperature for 1 h.

### 4.10. Immunofluorescence (IF) Assay

Cells were fixed with Immunol Staining Fix Solution (Beyotime, Shanghai, China) for 15 min at room temperature and washed three times for 5 min with Immunol Staining Wash Buffer (Beyotime, Shanghai, China). Immunostaining blocking solution (Beyotime, Shanghai, China) was added, and the mixture was incubated at room temperature for 2 h. Then, primary antibody (4 °C, overnight) and secondary antibody (room temperature, 1 h) were incubated with cells. Nuclei were stained with Hoechst. A confocal laser microscope was used to observe the cytoplasmic localization of the nucleus.

### 4.11. Cell Proliferation Assay

Cells (1 × 10^4^) were seeded into 96-well plates, vectors were transfected into cells and cell proliferation activity was detected using a Cell Counting Kit-8 (Beyotime, Shanghai, China). Briefly, at the indicated time points, the cells were incubated with 10 μL CCK-8 for 2 h at 37 °C. The absorbance was measured at 450 nm.

### 4.12. EdU Incorporation Assay

The cells were coated on slides and fixed in 4% paraformaldehyde for 15 min. The BeyoClick^TM^ EdU Cell Proliferation Kit with Alexa Fluor 555 (Beyotime, Shanghai, China) was used to detect cell proliferation. Cell proliferation was observed by fluorescence microscopy.

### 4.13. Cell Cycle and Apoptosis Analysis

Cell cycle and apoptosis were measured using the Cell Cycle and Apoptosis Analysis Kit (Beyotime, Shanghai, China) according to the manufacturer’s instructions. MSB1 cells were collected, 70% ice-cold ethanol was added, fixed at 4 °C overnight, and the supernatant was removed and then stained with propidium. DNA content was detected via flow cytometry, and cell cycle and apoptosis analyses were performed based on the distribution of DNA content.

### 4.14. RNA Sequencing and Differential Expression Analysis

Total RNA was extracted from three kinds of MSB1 cells which stably express circRUNX2.2, circRUNX2.2-ΔATG and pcl-ciR5, with two biological replicates for each condition. Briefly, the libraries were generated using the NEBNext Ultra Directional RNA Library Prep Kit for Illumina. Sequencing was performed on Illumina platforms at Novogene Bioinformatics Technology Co., Ltd. (Beijing, Shanghai, China), following the PE150 strategy. The clean data obtained after quality control were aligned to the reference genome using Hisat2. Subsequently, StringTie was employed for quantifying gene expression levels, and DESeq2 was utilized to conduct differential expression analysis. Gene ontology (GO) enrichment and KEGG pathway (http://www.genome.jp/kegg/, accessed on 1 October 2024) analysis were conducted using the bioinformatics online platform (https://www.omicshare.com/, accessed on 1 August 2024) and the generation of relevant statistical plots were conducted using the R (version 4.4.2).

### 4.15. Statistical Analysis

All statistical data were analyzed using SPSS software (version 16.0; SPSS, Chicago, IL, USA) and GraphPad Prism 8.3.0 (GraphPad, San Diego, CA, USA). Student’s *t*-test and one-way analysis of variance were used to analyze the statistical differences. Data are expressed as mean ± standard deviation (SD). Statistical significance was set at *p* < 0.05.

## Figures and Tables

**Figure 1 ijms-25-11486-f001:**
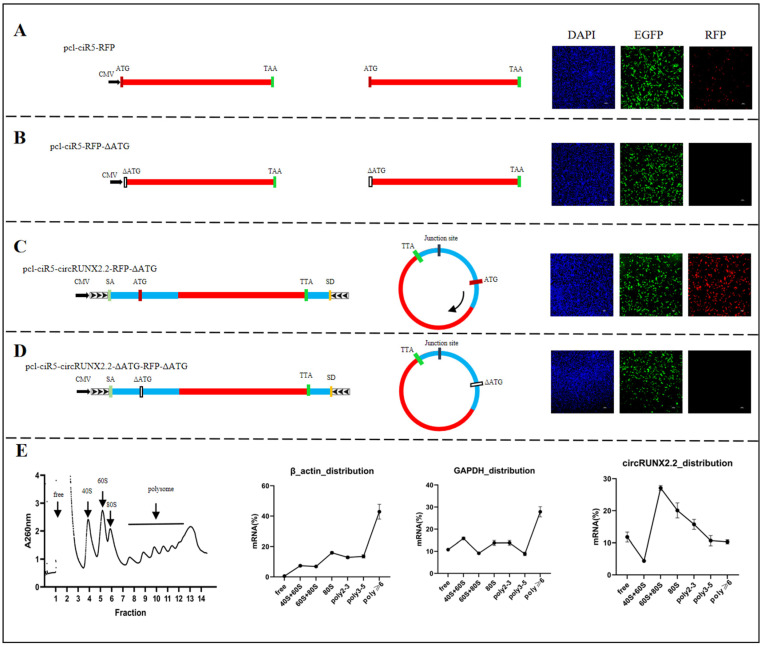
Translation capacity verification of circRUNX2.2. (**A**) Left panel: Schematic diagram for pcl-ciR5-RFP. RFP coding sequences were cloned into pcl-ciR5 vector to construct RFP positive expression vector. Middle panel: The theoretical transcript form. Right panel: RFP expression detected via fluorescence microscope after transfected into DF-1 cells for 48 h. (**B**) Left panel: Schematic diagram for pcl-ciR5-RFP-∆ATG. RFP coding sequences without the start codon ATG were cloned into pcl-ciR5 vector. Middle panel: The theoretical transcript form. Right panel: RFP expression detected via fluorescence microscope after transfected into DF-1 cells for 48 h. (**C**) Left panel: Schematic diagram for pcl-ciR5-circRUNX2.2-RFP-∆ATG. The RFP coding sequences without start codon ATG were inserted into the predicted ORF of circRUNX2.2, which cloned between splicing acceptor, splicing donor and side flanking repeat sequences (black arrows). Middle panel: The theoretical transcript form. Right panel: RFP expression detected via fluorescence microscope after transfected into DF-1 cells for 48 h. (**D**) Left panel: Schematic diagram for pcl-ciR5-circRUNX2.2-∆ATG-RFP-∆ATG. The RFP coding sequences without start codon ATG were inserted into the predicted ORF of circRUNX2.2 with start codon ATG deleted, which were cloned between splicing acceptor, splicing donor and side flanking repeat sequences (black arrows). Middle panel: The theoretical transcript form. Right panel: RFP expression detected via fluorescence microscope after transfected into DF-1 cells for 48 h. (**E**) cicRUNX2.2 was detected from polysome fractions. Polysome fractions were extracted from MSB1 cells by performing 10% to 50% sucrose gradient ultracentrifugation. circRUNX2.2 was detected via qPCR in the indicated fractions. GAPDH and β-actin served as the positive control. Scale bars = 100 µM, EGFP: A fluorescent tag on the vector (green), SA: Splicing acceptor, SD: Splicing donor. Nucleus stained with DAPI (blue).

**Figure 2 ijms-25-11486-f002:**
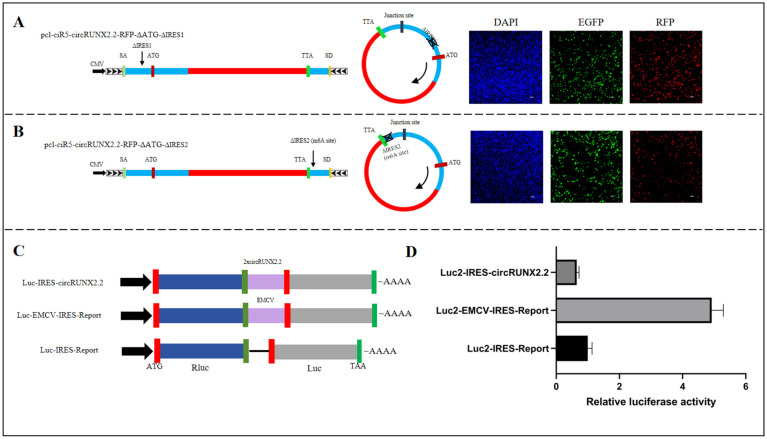
Analysis of the translation mechanism of circRUNX2.2. (**A**) Left panel: The RFP coding sequences without start codon ATG were inserted into the predicted ORF of circRUNX2.2 with the first predicted internal ribosomal entry site (IRES1) deleted, and the fusion sequences were cloned between splicing acceptor, splicing donor and side flanking repeat sequences (black arrows). Middle panel: The theoretical transcript form of the constructed vector. Right panel: RFP expression observed via fluorescence microscope after transfection of vector into DF-1 cells for 48 h. (**B**) Left panel: The RFP coding sequence without start codon was inserted into the predicted ORF with the second predicted IRES (IRES2) deletion of circRUNX2.2, and the fusion sequences were cloned between splicing acceptor, splicing donor and side flanking repeat sequences (black arrows). Middle panel: Theoretical transcription product of the constructed vector. Right panel: The red fluorescent protein expression observed via fluorescence microscope after transfection of vector into cells for 48 h. (**C**) Schematic diagram of the construction of a dual-luciferase reporter gene vector for detecting the IRES activity of circRUNX2.2. EMCV. Fragment of EMCV with IRES activity was used as a positive control. (**D**) Results of IRES activity detection. All values were shown as the mean ± SD. Scale bars = 100 µM, EGFP: A fluorescent tag on the vector (green), SA: splicing acceptor, SD: splicing donor. Nucleus stained with DAPI (blue).

**Figure 3 ijms-25-11486-f003:**
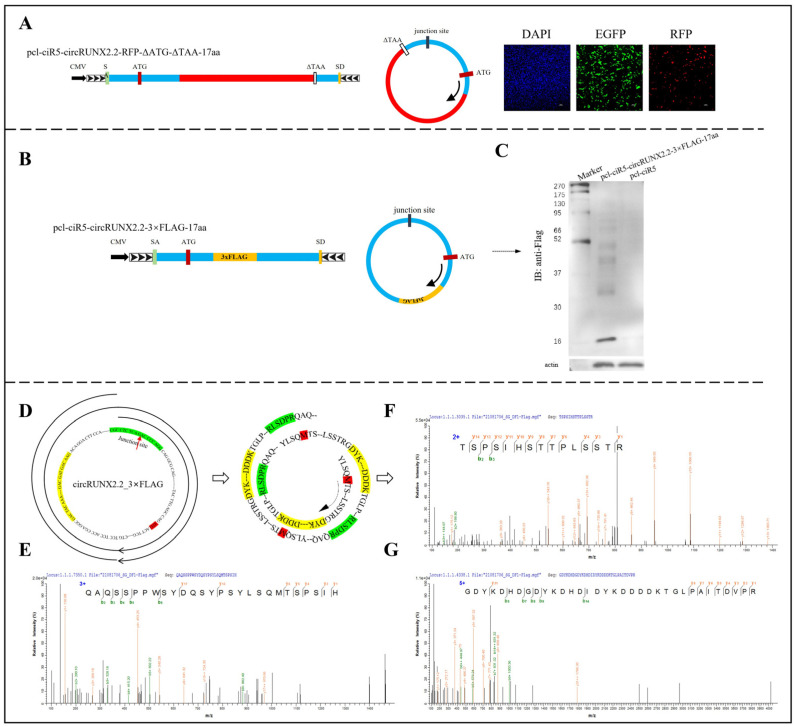
CircRUNX2.2 encoded small peptide in a rolling circle translation manner. (**A**) The RFP coding sequences without start codon ATG and stop codon TAA were inserted at the 17th amino acid site of the predicted ORF of circRUNX2.2, and the fusion sequence was cloned between splicing acceptor, splicing donor and side flanking repeat sequences (black arrows). Middle panel: Theoretical transcription product of the constructed vector. Right panel: The red fluorescent protein expression observed via fluorescence microscope after transfection of vector into cells for 48 h. (**B**) Left panel: 3×FLAG sequences was inserted at the 17th amino acid site of the predicted ORF of circRUNX2.2, and the fusion sequences were cloned between splicing acceptor, splicing donor and side flanking repeat sequences (black arrows). Right panel: Theoretical transcription product structure. (**C**) Western blot of fused protein. Western blot assay was performed to detect the translation products after transfection of vector into cells for 48 h. (**D**) Left panel: Schematic diagram of theoretical transcription product structure of circRUNX2.2_3×FLAG-17aa in rolling circle form. Right panel: Structure of the theoretical translation product of circRUNX2.2_3×FLAG-17aa in rolling circle manner. The start codon was marked in red, 3×FLAG sequence was marked in yellow, and junction region was marked in green. The red arrow pointed to the junction site. Black arrows represent transcription or translation direction. (**E**–**G**) CircRUNX2.2 and 3×FLAG fusion peptide was detected by mass spectrometry. Scale bars = 100 µM, EGFP: A fluorescent tag on the vector, SA: Splicing acceptor SD: Splicing donor. Nucleus stained with DAPI (blue).

**Figure 4 ijms-25-11486-f004:**
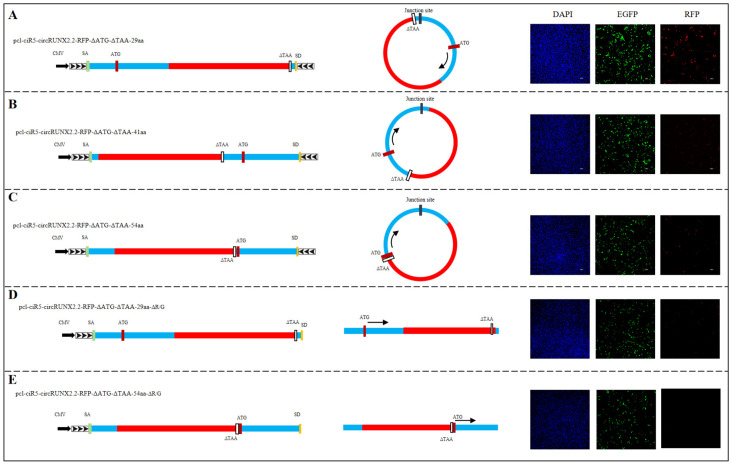
The translation efficiency of circRUNX2.2_RFP fusion protein. (**A**–**C**) Left panel: Schematic diagrams for constructed vectors with RFP inserted in different positions. The RFP coding sequences without start codon ATG and stop codon TAA were inserted at the 29th, 41st, 54th amino acid site of the predicted ORF of circRUNX2.2, respectively, and the fusion sequences were cloned between splicing acceptor, splicing donor and side flanking repeat sequences (black arrows) to form pcl-ciR5-circRUNX2.2-RFP-∆ATG-∆TAA-29aa, pcl-ciR5-circRUNX2.2-RFP-∆ATG-∆TAA-41aa and pcl-ciR5-circRUNX2.2-RFP-∆ATG-∆TAA-54aa vectors. Middle panel: Theoretical transcription product of the constructed vector. Right panel: The red fluorescent protein expression observed via fluorescence microscope after transfection of vectors into cells for 48 h. (**D**,**E**) Left panel: Schematic diagram for constructed linear vectors with the deletion of downstream sequence mediating loop formation. The vectors pcl-ciR5-circRUNX2.2-RFP-ΔATG-ΔTAA-29aa-ΔR/G and pcl-ciR5-circRUNX2.2-RFP-ΔATG-ΔTAA-54aa-ΔR/G were constructed by deleting the downstream sequence mediating loop formation to destroy cyclization. Middle panel: Theoretical transcription product of the constructed vector. Right panel: The red fluorescent protein expression observed via fluorescence microscope after transfection of vector into cells for 48 h. Scale bars = 100 µM, EGFP: A fluorescent tag on the vector (green). Nucleus stained with DAPI (blue).

**Figure 5 ijms-25-11486-f005:**
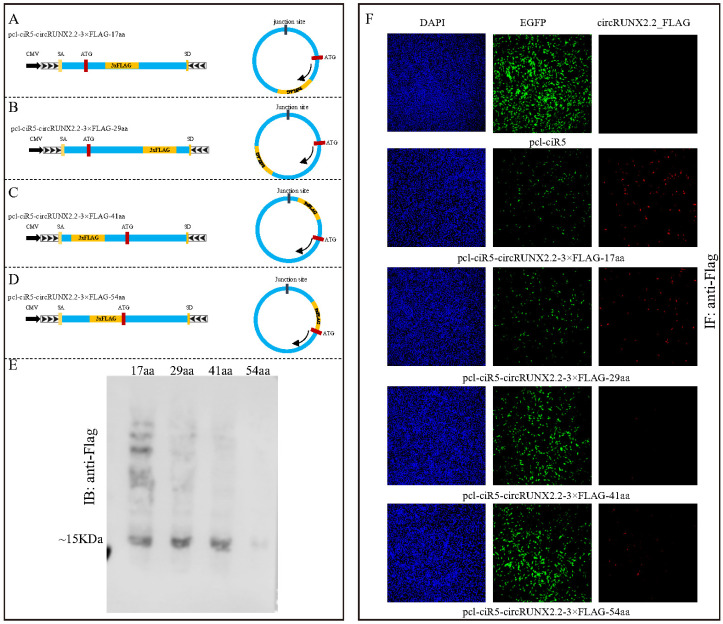
The translation efficiency of circRUNX2.2_FLAG fusion protein. (**A**–**D**) Left panel: Schematic diagrams for constructed vectors with 3×FLAG inserted in different positions. 3×FLAG sequences were inserted at the 17th, 29th, 41st, 54th amino acid site of the predicted ORF of circRUNX2.2, and the fusion sequences were cloned between splicing acceptor, splicing donor and side flanking repeat sequences (black arrows) to construct pcl-ciR5-circRUNX2.2-3×FLAG-17aa, -29aa, -41aa and -54aa vectors. Right panel: Theoretical transcription product of the constructed vector. (**E**) Western blot assay detecting the translation products of circRUNX2.2_FLAG fusion protein (17aa, 29aa, 41aa, 54aa) with anti-FLAG. (**F**) Fused circRUNX2.2_FLAG (17aa, 29aa, 41aa, 54aa) translation products examined by immunofluorescence with anti-FLAG. pcl-ciR5 vector was the negative control. The Cy5 (red) fluorescence was observed via fluorescence microscope. Scale bars = 100 µM, EGFP: A fluorescent tag on the vector (green). Nucleus stained with DAPI (blue).

**Figure 6 ijms-25-11486-f006:**
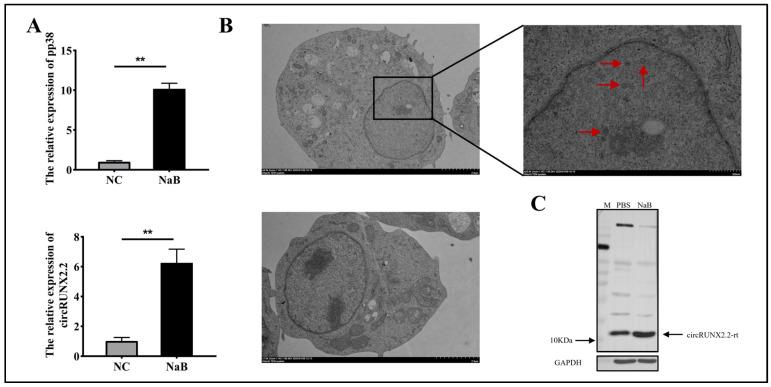
circRUNX2.2 was induced by MDV reactivation. (**A**) The relative expression of circRUNX2.2 and PP38 gene in MSB1 cells treated with NaB. (**B**) Electron micrographs of MDV particles. MSB1 cells were treated with NaB and electron microscopy was used to observe MDV virus particles. The upper panel was MSB1 treated with NaB1, and the lower panel was MSB1 treated with PBS as a control. The red arrows in the figure represented MDV virus particles, which appear hexagonal. (**C**) Western blot assay of endogenous circRUNX2.2. The endogenous circRUNX2.2-rt was detected after MSB1 cells treated with NaB and negative control. Small peptides with different molecular weight sizes encoded by circRUNX2.2 were identified. M marker, NC: negative control, NaB: NaB treatment. The data were represented as mean ± SD, ** *p* < 0.01. Scale bars = 130 µM.

**Figure 7 ijms-25-11486-f007:**
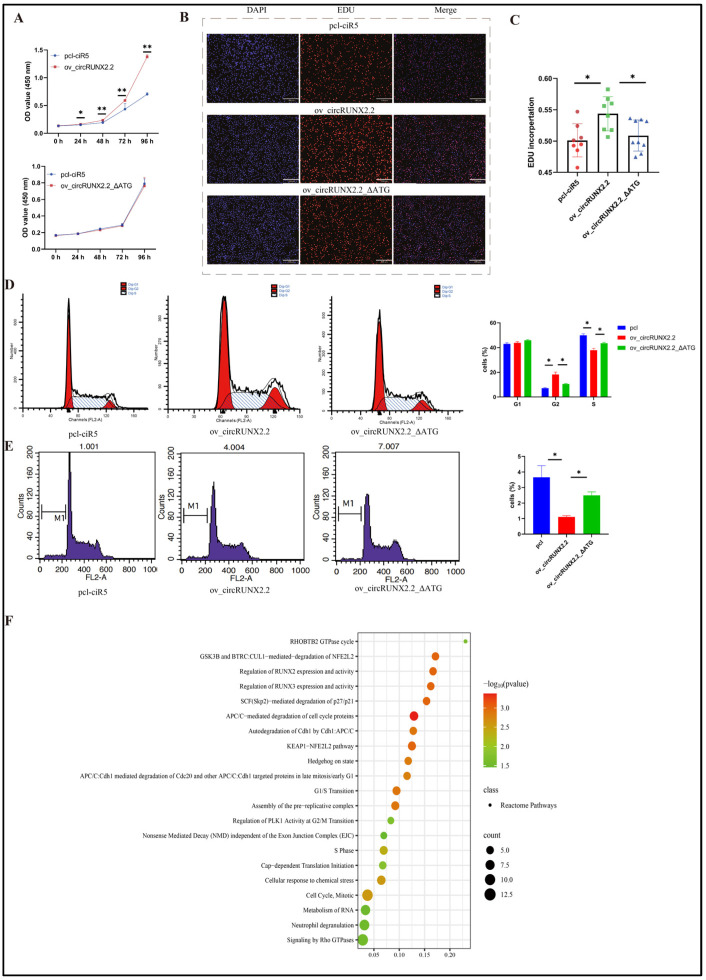
circRUNX2.2 promoted MSB1 development. (**A**) CCK-8 assay was performed to evaluate cell viability after MSB1 cells transfected with vectors for 48 h. Upper panel: CCK-8 assay results after overexpression of circRUNX2.2 and pcl-ciR5. Lower panel: CCK-8 assay results after overexpression of circRUNX2.2-∆ATG and pcl-ciR5. (**B**) The EdU and DAPI staining analysis of MSB1 cells after pcl-ciR5, circRUNX2.2 circRUNX2.2-∆ATG vectors transfection. The EdU and DAPI staining analysis was conducted to detect MSB1 cell proliferation ability between overexpression of circRUNX2.2 and of circRUNX2.2-∆ATG compared to empty vector. (**C**) Statistics for the proportion of proliferating cells in EdU staining analysis. The cell cycle (**D**) and apoptosis (**E**) assay of MSB1 transfected with pcl-ciR5, circRUNX2.2 and circRUNX2.2-∆ATG vectors. Cells were stained with propidium, DNA content was analyzed via flow cytometry, cell cycle and apoptosis analysis was calculated based on the distribution of DNA content. (**F**) Analysis of the circRUNX2.2-rt interaction protein enrichment pathway. The 168 proteins detected by immunoprecipitation combined with mass spectrometry were submitted to the STRING website for analysis, and the Reactome channel was used to analyze the enriched pathways. The data were represented as mean ± SD, * *p* < 0.05, ** *p* < 0.01. Scale bars = 130 µM.

## Data Availability

All data are shown in the paper in figures, tables or Appendix A. RNA-seq data are available at NCBI Gene Expression Omnibus (PRJNA1070428).

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
