# Peer review of "Rolling-Translated circRUNX2.2 Promotes Lymphoma Cell Proliferation and Cycle Transition in Marek’s Disease Model"

_ijms, 2024, doi:10.3390/ijms252111486_

Round 1
Reviewer 1 Report
Comments and Suggestions for Authors
Here the authors investigate the role of circRUNX2.2 in chicken Marek’s disease lymphoma, finding it significantly upregulated in MDV-infected tumorous spleens.
They show that circRUNX2.2 translates small peptides in a rolling circle manner, independent of IRES or m6A-mediated mechanisms. They also demonstrate that the translated peptides promote cell proliferation and inhibit apoptosis in lymphoma cells, suggesting a potential oncogenic role. These findings provide new insights into the translation mechanisms of circRNAs and contribute to understanding lymphoma mechanisms and may aid in developing new therapeutic strategies.
Major Point
- The study describes findings in only one cellular model. It would be important to test the effects of circRUNX2.2 peptides in various lymphoma cell lines to determine if the oncogenic properties are specific to certain cell types. If not possible, the researchers should justify why they are using only one cell model.
Minor points
- A limitation paragraph should be added, as the study focuses only on a specific model (chicken Marek’s disease), which may limit broader applicability.
- The authors could also add a paragraph focusing on future directions for this research topic, such as investigating the signaling pathways and molecular interactions influenced by circRUNX2.2 peptides to understand the underlying mechanisms of their oncogenic activity.
- The authors should also discuss how this research is therapeutically relevant, by explaining if it would be possible to develop and test small molecules or antibodies that specifically target circRUNX2.2 peptides to evaluate their potential as therapeutic agents.
Author Response
Here the authors investigate the role of circRUNX2.2 in chicken Marek’s disease lymphoma, finding it significantly upregulated in MDV-infected tumorous spleens.
They show that circRUNX2.2 translates small peptides in a rolling circle manner, independent of IRES or m6A-mediated mechanisms. They also demonstrate that the translated peptides promote cell proliferation and inhibit apoptosis in lymphoma cells, suggesting a potential oncogenic role. These findings provide new insights into the translation mechanisms of circRNAs and contribute to understanding lymphoma mechanisms and may aid in developing new therapeutic strategies.
Major Point
Comments 1: The study describes findings in only one cellular model. It would be important to test the effects of circRUNX2.2 peptides in various lymphoma cell lines to determine if the oncogenic properties are specific to certain cell types. If not possible, the researchers should justify why they are using only one cell model.
Response 1:
Thank you very much for your thorough review of our research work and providing valuable suggestions. It is important to test the effects of circRUNX2.2 peptides in various lymphoma cell lines. However, except MSB1, the only cell model that we can get is DT40 chicken B cell line, derived from an avian leukosis virus (ALV)-induced bursal lymphoma, which was used to detect the effect of circRUNX2.2 peptide on its proliferation. The EdU labeling results showed the number of EdU labeled cells in the circRUNX2.2 overexpression group was more than those in the two control groups, circRUNX2.2_ΔATG or pcl-ciR5 overexpression group (Fig. 1 A-B) when circRUNX2.2 and circRUNX2.2_ΔATG were overexpressed (about 20 times) (Fig. 1 C). These results suggested that circRUNX2.2 peptide had promotion effect on the proliferation of DT40 cells, and this promotion effect was mitigated when the circRUNX2.2 open reading frame was destroyed.

Fig. 1 The effect of circRUNX2.2 on DT40 cell proliferation
A) The overexpression of circRUNX2.2 in DT40 cells. B) Statistical results of EDU labeled cells. C) Staining results of EDU labeled cells
Based on the above results, we believe that circRUNX2.2 small peptide may not only regulate the growth of Marek's disease lymphoma cells, but also have a certain impact on the growth of other lymphoma cells.
Minor points
Comments 2: A limitation paragraph should be added, as the study focuses only on a specific model (chicken Marek’s disease), which may limit broader applicability.
Response 2:
Thank you for your advice. We have supplemented one paragraph about the conservation of circRUNX2.2 across species. If this gene exhibited high conservation across species, our discoveries may hold much wider implications since the significant role of circRUNX2.2 in the chicken model has been investigated. Thus, we compared chicken circRUNX2.2 sequences across multiple species, including human, rat, cat and pig, and the results showed that the similarity within each sequence was high (>80%). The similarity within each predicted peptide sequence among species showed that only one amino acid difference between rat and the other four species, and one amino acid difference between chicken and the other four species. We also validated the peptide encoded by human circRUNX2.2 via WB and IF assay. The across species conservation analysis revealed that the potentially similar biological function of circRUNX2.2 across species which will expand its potential applications. Please see lines 200-217 and Supplementary Fig. 1.
Comments 3: The authors could also add a paragraph focusing on future directions for this research topic, such as investigating the signaling pathways and molecular interactions influenced by circRUNX2.2 peptides to understand the underlying mechanisms of their oncogenic activity.
Response 3:
Thank you for your suggestion. To understand the underlying mechanism of circRUNX2.2 peptide in lymphoma cell development, RNA-seq in circRUNX2.2 overexpression MSB-1 cell was conducted. Totally, 1533, 1364, and 1336 differentially expressed genes were identified in three comparison groups (circRUNX2.2_vs._pcl-ciR5, circRUNX2.2-â–³ATG_vs._pcl-ciR5, and circRUNX2.2_vs._circRUNX2.2-â–³ATG), respectively. Gene expression trends analysis revealed that 40 DEGs were upregulated and 77 DEGs were downregulated DEGs in circRUNX2.2 overexpression group relative to circRUNX2.2-â–³ATG and pcl-ciR5 groups. The upregulated genes were enriched in protease activity and pathways related to cell cycle, autophagy, apoptosis, and necrosis, with ubiquitin specific peptidase 7 (USP7) being significantly upregulated. The downregulated genes were enriched in ubiquitin-associated protease activity and pathways related to autophagy, necrosis, and viral diseases such as herpes simplex virus 1 infection. Overall, the results suggested that circRUNX2.2 promoted lymphoma cell development through regulation of protein ubiquitination/deubiquitination, which needs to be verified in further study. Please see Supplementary Fig. 4 and lines 398-423 and 522-538.
Comments 4: The authors should also discuss how this research is therapeutically relevant, by explaining if it would be possible to develop and test small molecules or antibodies that specifically target circRUNX2.2 peptides to evaluate their potential as therapeutic agents.
Response 4:
Thank you for your suggestion. We have additionally explored the possibility of using circRUNX2.2 as a therapeutic target in the discussion section. Currently, several therapeutic strategies were explored with the aim to overexpress or knockdown circRNAs. Considering that circRUNX2.2 functioned as an oncogene in MD, knocking down its expression was the ideal therapeutic strategy. Since the high GC content at the junction region of circRUNX2.2, it was difficult to design an effective siRNA to knock it down, the CRISPR/Cas9-mediated circRNA knockout or knockdown and the CRISPR/Cas13-mediated circRNA knockdown technologies may contribute to the explore the therapeutics targeting circRUNX2.2. We had added this discussion. Please see lines 514-521.
Reviewer 2 Report
Comments and Suggestions for Authors
Dear authors of “Rolling-translated circRUNX2.2 presents an oncogene role in chicken Marek’s disease lymphoma model”,
Thanks for your contribution to this field. This is an interesting article aimed at identifying new rôle for circRUNX2.2 in lymphoma desease.
The manuscript is well written. Research is well organized, and the obtained results are convincing. The findings are of interest for circulating RNA. They are well-discussed regarding the actual literature.
Nevertheless, the input for lymphoma research is less evidenced. The claimed oncogen role is only supported by proliferation and cell cycle analyses.
It will be worth to add information on the impact in lymphoma to improve the manusript such as the induced signalling pathway, the effects on glycolytic and oxidative metabolism, or on cell migration.
Looking forward to seeing your modifications,
All the best,
Author Response
Dear authors of “Rolling-translated circRUNX2.2 presents an oncogene role in chicken Marek’s disease lymphoma model”, Thanks for your contribution to this field. This is an interesting article aimed at identifying new role for circRUNX2.2 in lymphoma disease. The manuscript is well written. Research is well organized, and the obtained results are convincing. The findings are of interest for circulating RNA. They are well-discussed regarding the actual literature. Nevertheless, the input for lymphoma research is less evidenced. The claimed oncogene role is only supported by proliferation and cell cycle analyses. It will be worth to add information on the impact in lymphoma to improve the manusript such as the induced signalling pathway, the effects on glycolytic and oxidative metabolism, or on cell migration.
Looking forward to seeing your modifications,
All the best,
Comments 1: Nevertheless, the input for lymphoma research is less evidenced. The claimed oncogene role is only supported by proliferation and cell cycle analyses.
It will be worth to add information on the impact in lymphoma to improve the manusript such as the induced signalling pathway, the effects on glycolytic and oxidative metabolism, or on cell migration.
Response 1:
Thank you for your valuable review. We have conducted RNA sequencing (RNA-seq) in MSB-1 cells among circRUNX2.2 overexpression, circRUNX2.2-â–³ATG overexpression, and pcl-ciR5 overexpression. Results showed that 1533, 1364, and 1336 differentially expressed genes (DEGs) were identified in three comparison groups (circRUNX2.2_vs._pcl-ciR5, circRUNX2.2-â–³ATG_vs._pcl-ciR5, and circRUNX2.2_vs._circRUNX2.2-â–³ATG), respectively. Gene expression trends analysis revealed that 40 DEGs were upregulated and 77 DEGs were downregulated in circRUNX2.2 overexpression group relative to circRUNX2.2-â–³ATG and pcl-ciR5 groups. The upregulated genes were enriched in protease activity and pathways related to cell cycle, autophagy, apoptosis, and necrosis, with ubiquitin specific peptidase 7 (USP7) being significantly upregulated. The downregulated genes were enriched in ubiquitin-associated protease activity and pathways related to autophagy, necrosis, and viral diseases such as herpes simplex virus 1 infection. Overall, the results suggest that circRUNX2.2 promotes lymphoma cell development through regulation of protein ubiquitination/deubiquitination, which needs to be verified in further study. Please see Supplementary Fig. 4 and lines 398-423 and 522-538.
Round 2
Reviewer 1 Report
Comments and Suggestions for Authors
The authors addressed all the issues.
Author Response
Thank you for confirming that we have successfully addressed all the issues raised during the review process.
Reviewer 2 Report
Comments and Suggestions for Authors
Dear authors of “Rolling-translated circRUNX2.2 presents an oncogene role in chicken Marek’s disease lymphoma model”,
Thanks for your answer to my comments.
Indeed, the problem is still ongoing. In your title and manuscript, you claim that Rolling-translated circRUNX2.2 presents an oncogene, but any experimental evidence is present to support your finding. As you write, all are suggestions or speculations.
That is why I was suggesting adding experimental evidence on the induced signaling pathway, the effects on glycolytic and oxidative metabolism, or cell migration.
As I understand it, you prefer keeping those data for another publication, which can make sense. In such a case, it is better to adjust your title to something like “Rolling-translated circRUNX2.2 dysregulates oncogenes in chicken Marek’s disease lymphoma model”.
Regards,
Author Response
Thank you for your valuable suggestion. To more accurately reflect the content and conclusions of our current study, we agree to adjust the title of the paper to "Rolling-Translated circRUNX2.2 promote cell proliferation and cycle transition in Marek's Disease Lymphoma Model".
Round 3
Reviewer 2 Report
Comments and Suggestions for Authors
Dear authors of “Rolling-translated circRUNX2.2 presents an oncogene role in chicken Marek’s disease lymphoma model”,
Thanks for the change in your title which reflect my comments.
Now the manuscript is suitable for publication.
Congratulations,